# Applying Machine Learning Algorithms for Kidney Disease Diagnosis

## Abstract

Kidney disease is a 'silent killer'; it usually develops over time, and many people do not know they have it until it is very far along. Chronic Kidney Disease(CKD) represents a major public health problem in both developed and developing countries. In this research, three different classification algorithms have been used in order to evaluate the occurrence of ckd on a dataset collected from the UCI Repository which holds 400 samples with 25 attributes. By filtering out the top 14 features from the 24 input variables which show high score of dependability, the experimental results manifest that Multi Layer Perceptron (MLP) works best for both normalized and unnormalized features yielding an accuracy of 100%.

## Introduction

Kidney is the most silent, yet one of the most vital organs serving two important functions in human beings: (i) removal of harmful substances from the blood, and (ii) retention of the useful components in a proper balance. Chronic kidney disease (CKD) occurs when a disease or condition impairs kidney function, causing kidney damage to worsen over several months or years. In Ethiopia, it is estimated that 5-10 percent of adults are affected by kidney disease of various levels. In addition, around 7000 people, most of them below the age of 55, die annually due to chronic kidney disease [1]. One of the major factors that are escalating the problem is limited access to kidney medical care and also increased risk factors such as high blood pressure and diabetes mellitus. However, if detected early, CKD could be treated, thereby reducing complications of the health problem as well as the medical process and healthcare costs [2].

## The Problem

In the case of Ethiopia, there's a shortage of health facilities, trained doctors and clinicians. The burden of kidney failure is concealed behind statistics that reflect only the number of people treated, not the number who die of kidney failure. And with only a total of 78 radiographers and 20 radiologists in Addis Ababa's public hospitals, and no radiologist in three of them, the problem of identifying kidney disease is exacerbated. It is known that proper diagnosis of kidney disease is greatly impacted by the human expertise (radiologist) and the ability to obtain all the necessary attributes and produce the right decision by infering from those inputs. When treating kidney disease, dialysis or transplantation may become necessary. while Spain and Poland have more than 200 dialysis centers, Ethiopia with a population twice that of Poland and Spain combined has only two dialysis centers and two nephrologists and even if the centers are present, the cost of dialysis and transplantation is extremely prohibitive for the general population. Therefore this research examines feature selection and classification algorithms in order to identify which top features give strength for predicting ckd which in turn means, by obtaining these few attributes and using the right algorithm for classification, it's possible to carry out the diagnosis and use the system as assistant for the physicians.

## Methods

The data for this research is obtained from UCI data repository. It includes 400 sample data with 25 attributes. There are 24 input variables and one output variable/target. Among the 24 attributes, 14 are continuous or numerical and the rest are nominal. The output is labeled as either: ckd or notckd. Various approaches have been taken to identify kidney disease before [3-6]. In this paper, first preprocessing is done by filling missing gaps in the data through propagating the non-NaN values forward with a limit of 1

and proceeded by backward filling algorithm along the column. Then Using Machine Learning PySpark-StringIndexer, the categorical feature taken as input column is transformed into an indexed numerical feature. After this, normalization is carried out on all the features in order to make training less sensitive to the scale of features. The next step is feature selection, and for this selectKBest is used with a score function chi2 to give the highest scores which means the feature is non-randomly related to the target, and so likely to provide important information. Through these 14 features with highest score are taken as input for the classification stage. Data is split into training set and testing set. 70 % of the data is used for training the model and the rest for testing. Finally, three different machine learning methods namely: K-nearest Neighbour (KNN), Support vector machine (SVM) and multi-layer perceptron (MLP) are applied on both original and normalized data set.

## Results

Table1 shows the top 14 features which give high score value and therefore highly correlated with the target value and Table2 shows the performance of the classification algorithms given the top 14 features as input and we can see that the performance of KNN has dropped from 99.17% to 70% when using unnormalized data, SVM a little bit but as for MLP, it has retained its performance for both cases with an accuracy of 100%. The developed system is to be used: 1- for the purpose of assisting doctors to predict disease properly, 2- to focus on the selected attributes when collecting data from local hospitals and 3- to be integrated with the ongoing project with the intention of combining the selected attributes with image processing techniques so as to achieve further classification on the type and stage of kidney disease which in turn helps in taking effective measures accordingly. In a country where medical assistance, quality and service at a reasonable price is hard to find, chronic diseases like kidney failure and its unending trips to the hospital for dialysis is unbearable. Therefore, having a system which identifies the kidney disease early and help in decision making is a necessity.

Table 1**. Top fourteen (14) selected features**

| Selected Features | | |
|---|---|---|
| Specific gravity (sg) | Serum Creatinine (sc) | White Blood Cell Count (wbcc) |
| Potassium (pot) | Blood Glucose Random (bgr) | Red Blood Cell Count (rbcc) |
| Sodium (sod) | Blood Urea (bu) | Albumin (al) |
| Blood Pressure (bp) | Hemoglobin (hemo) | Age (age) |
| Sugar (su) | Packed Cell   Volume (pcv) | |

Table 2. **Predictive accuracy of classification algorithms**

| Classification algorithms | Accuracy (%) | |
|---|---|---|
| | With Normalized selected features | With unnormalized selected Features |
| **KNN** | 99.17 | 70 |
| **SVM** | 100 | 99.17 |
| **MLP** | 100 | 100 |

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
