# OpenReview forum: "Applying Machine Learning Algorithms for Kidney Disease Diagnosis"
_MIDL.io/2020/Conference — Submitted to MIDL 2020_

### Official Review · AnonReviewer1 · 2020-02-21
**Respectable Project, but no scientific Impact**

**Rating:** 1
**Confidence:** 4

**Review:**

In this work, the authors tackle the problem of automatic kidney disease diagnosis using machine learning. The main motivation is the insufficient healthcare coverage in Ethiopia.
They present a study on using three standard machine learning algorithms on a public dataset after a feature selection and normalization.
Even though the overall aim of the project is remarkable, the authors fail to deliver anything new and hence, I would highly question the scientific impact made in this work! There are numerous studies on exactly the same dataset where the same algorithms were applied!
Additionally, the description of the methods is very poor. The feature selection method is not explained, only a name to a python function is given. I cannot find a single word on the parameters of the machine learning methods, for example the MLP architecture, the SVM Kernel or the number of neighbors in KNN. The result that the methods perform better on normalized features is quite expectable and, from my point of view, not a significant contribution.
On top of this, the work might be out of scope for the MIDL as neither medical images nor deep learning is part of the work.
Hence, I would vote to reject the work in its current form. However, I want to highly encourage the authors to keep on tackling the very important issue of solving the problem of insufficient medical aid coverage using modern technology!

---

### Official Review · AnonReviewer3 · 2020-03-10
**data is not medical imaging, approach is not deep learning, not within the scope of MIDL---lack of innovation/novelty/....**

**Rating:** 1
**Confidence:** 5

**Review:**

Authors used three different machine learning classifiers for CKD diagnosis (binary decision). Authors used publicly available UCI data sets (400 sample data with 25 attributes).  MLP, KNN, and SVM were used and compared, MLP was decided to give better results, as an outcome of the study.

-- the paper does not show any innovation, no technical novelty, the use of known 3 classifiers on a known data set, nothing really is new.
-- table 2 does not provide any sensitivity and specificity, accuracy itself is a not a good (and enough) metric to define the success of the algorithm
-- Correlation does not mean causation; hence, selected features mayn't be really meaningful
-- I am not sure if the paper is within the scope of there MIDL which means "medical imaging" with deep learning, where I cant see any imaging but only some clinical variables to be used as features.

---

### Official Review · AnonReviewer2 · 2020-03-10
**This paper aims to use machine learning algorithm to identify Chronic Kidney Disease**

**Rating:** 2
**Confidence:** 3

**Review:**

1. The paper is well written but I have a concern that the author’s contribution is very low. There are many other papers that have used the same data and the same algorithms and reported the same results.
2. There are other criteria like AUC and sensitivity that have not been reported in this short paper.

---

### Official Review · AnonReviewer4 · 2020-03-17
**Direct analysis of existing data resource**

**Rating:** 1
**Confidence:** 4

**Review:**

This paper investigates machine learning on the UCI chronic kidney disease repository. First, preprocessing and data clearing are briefly described and included as important steps in the results, yet the novelty and innovation of this data cleaning step is unclear. Second, baseline algorithms (SVM, MLP, KNN) are applied to the cleaned and raw data. It is unclear why one would want to machine learn on data with known problems. Hence, the creativity / innovation of this approach is unclear. Overall, the method makes sense, but the contributions are unclear.

---

### Meta-Review · Area_Chair1 · 2020-04-06
**MetaReview of Paper21 by AreaChair1**

**Rating:** 1

**Metareview:**

This paper applies standard machine learning classifiers (SVM, MLP, kNN) to a non-imaging dataset predicting kidney disease. While the reviewers agree that the analysis is sound, the novelty is very limited and the relevance for MIDL is low given that the paper does not use medical imaging data.

**Paper Type:**

validation/application paper

---

### Decision · Program_Chairs · 2020-04-11

Reject